# Shear Bond Performance at Interface of Concrete-Filled Steel Tube after Freeze–Thaw Cycles

**DOI:** 10.3390/ma15207233

**Published:** 2022-10-17

**Authors:** Bo Xu, Yongjian Liu

**Affiliations:** 1Department of Bridge Engineering, School of Highway, Chang’an University, Xi’an 710064, China; 2Department of Civil Engineering, School of Civil Engineering, Ordos Institute of Technology, No. 1 Ordos Street, Ordos 017000, China

**Keywords:** shear bond performance, steel tube section type, concrete strength, freeze–thaw cycles, load–slip

## Abstract

The shear bond performance between steel tube and concrete is the basis of a synergistic effect between two materials. When a concrete-filled steel tube structure is damaged by freeze and thaw cycles, the shear bond performance will deteriorate, which will inevitably affect the safety of the structure. In this paper, the effects of the numbers of freeze–thaw cycles, section type of steel tube, and concrete strength on the failure mode, bond strength, and peak slip of concrete-filled steel tube are investigated. In addition, the load–slip curve, the shear bond strength calculation formula, and the peak slip calculation formula are obtained. The results show that the shear bond performance decreases with the increase in freeze and thaw cycles; after 20 freeze and thaw cycles, the shear bond strength of the round steel tube specimen with 30 MPa decreases by 33.33%, while that of peak slip increases by 11.49%; the shear bond strength increases with the increase in concrete strength, while the degree of decrease in shear bond strength after freeze–thaw cycles is reduced; the shear bond strength of the round steel tube push-out specimen is higher than that of square steel tube push-out specimen, and after 20 freeze and thaw cycles, the shear bond strength of the round steel tube push-out specimen with 30 MPa/45 MPa is 0.86 MPa/1.62 MPa, while that of square steel tube push-out specimen with 30 MPa/45 MPa is 0.40 MPa/0.50 MPa.

## 1. Introduction

Concrete-filled steel tube is a steel–concrete composite member formed by filling core concrete inside the steel tube [1]. Through the synergistic effect of steel tube and concrete, the shortcomings of the two materials forming structural components alone are overcome [2]. The structure makes full use of the good tensile properties of steel and compressive properties of concrete, so it is widely used in large-span bridge structures, super high-rise frame structures, and other practical engineering [3]. Concrete-filled steel tube is mainly divided into a rectangular section of concrete-filled steel tube and a circular section of concrete-filled steel tube according to the different cross-section forms [4,5]. The interfacial bonding stress of concrete-filled steel tube is composed of three parts: the chemical adhesion force, the mechanical bite force and the interface friction force. The chemical adhesion force is the adsorption force between the core concrete and the inner surface of the steel tube in the process of setting and hardening, which is related to the mix proportions of concrete; the unevenness of concrete embedded in stainless steel tube will generate mechanical bite force, and the mechanical bite force is related to inner surface roughness of the steel tube and the shear strength of concrete; when the concrete and the tube slip relative to each other, the interface friction force is generated, and the interface friction force is related to the steel tube restraint and friction coefficient of steel–concrete interface.

In cold areas, the concrete structure will suffer from freeze–thaw damage [6], which will degrade the concrete structure from the outside to the inside [7] and affect the bond performance between the steel pipe and the concrete [8]. In the actual working conditions, the steel tube and the concrete in the concrete-filled steel tube are often in different loading states, so the load is transmitted through the interface between the steel pipe and the concrete [9]. Therefore, the bond performance of the steel pipe and the concrete is the synergistic force between the two materials [10].

The height-to-diameter ratio [11], the diameter-to-thickness ratio [12], the concrete strength [13], cross-sectional dimension [14], the concrete age, etc., all had different degrees of influence on the bond performance between the steel tube and the concrete in concrete-filled steel tubes [15]. Liu et al. [12] investigated the bond performance between the round steel tube (square steel tube) and the concrete in concrete-filled steel tubes and found that the bond–slip curve shape of round steel tube concrete and square steel tube concrete was similar. The bond strength between the steel tube and the concrete increased with the increase in the height-to-diameter ratio of the steel tube; the bond strength decreased with the increase in the diameter-to-thickness ratio, while it increased slightly with the increase in concrete age. Xu et al. [13] found the larger the water–cement ratio of the core concrete, the smaller the interfacial bond strength between the steel tube and the concrete. Tao et al. [14] pointed out that the bond strength between the steel tube and the concrete decreased greatly with the increase in section size and concrete age, so the bond strength given by the specification was not safe enough. Nguyen et al. [16] investigated the interface between steel and concrete in concrete-filled steel tubes through the nonlinear finite element analyses method. Chen et al. [17] investigated the interfacial bond behavior of high strength concrete-filled steel tube after exposure to elevated temperatures and cooled by a fire hydrant, and the results show that the load–slip curves of the loaded end and the free end of the specimen were similar and could be divided into three typical curves; the strain on the outer surface of the steel tube was exponentially distributed with the distance from the loading end.

The freeze–thaw cycle has an extremely bad effect on concrete structures, and many scholars have studied the effect of freeze–thaw cycles on concrete-filled steel tube structures [18,19]. The research on concrete-filled steel tube structures in a freeze–thaw environment mainly focuses on the performance of concrete-filled steel tube short columns [20,21]. Yang et al. [22] conducted an experimental study on the axial compression of short columns of square and circular concrete-filled steel tubes under a freeze–thaw environment and found that the elastic modulus of the specimen decreased with the increase in the number of freeze–thaw cycles, and the bearing capacity of the concrete-filled steel tube short columns decreased, which is consistent with the conclusion obtained by Wang [23]. Li et al. [24] found that the effect of freeze–thaw cycles on the load–slip curve of concrete-filled steel tube columns was not obvious, but the interfacial bond performance of concrete-filled steel tube columns showed a downward trend, where the bond strength continued to decline, and the interface slip continued to increase. Shen et al. [25] point out that the initial stiffness of concrete-filled steel tube columns increased with the increase in concrete strength grade, while the freeze–thaw cycles have little effect on the axial compressive bearing capacity of concrete-filled steel tube columns.

However, there are few studies on shear bond performance at the interface of concrete-filled steel tubes with different sections after freeze–thaw cycles. In addition, few studies have considered the effect of concrete strength on shear bond performance at the interface of concrete-filled steel tubes after freeze–thaw cycles. The purpose of this paper is to investigate the effects of the numbers of freeze–thaw cycles, section types of steel tube, and concrete strength on the failure mode, bond strength, and peak slip of concrete-filled steel tube. It can promote the application of concrete-filled steel tubes in cold areas.

## 2. Experiment

### 2.1. Materials and Mix Proportions

In this paper, the natural crushed stone was used as coarse aggregates, the particle range was 5–20 mm, the water absorption was 1.4%, and the crush indicator was 4.1% [26]. The natural river sand was used as fine aggregates, the fineness modulus was 2.7, the particle diameter was less than 4.75, and the mud content was 1.9%. The cement type was P.O 42.5, the performance of cement was listed in Table 1 [27]. The steel tubes were divided into a round steel tube and a square steel tube, the steel type was Q235, as shown in Figure 1. The yield strength of Q235 steel was 307 MPa, the ultimate strength was 377 MPa, and the elongation was 26.62%. The thermal conductivity of the steel tube was 22 W/m·K. The mixed proportions of concrete in this paper are listed in Table 2 [28]. The target 28-day compressive strength of concrete in this paper was 30 MPa/45 MPa, and the measured 28-day compressive strength after maintaining in a standard curing box was 28 MPa/46 MPa. The thermal conductivity of concrete was 1.28 W/m·K.

### 2.2. Specimens

For the study, thirty-six push-out specimens were cast to investigate the shear bond performance at the interface of a concrete-filled steel tube, as shown in Figure 2. The height-to-diameter ratio (L/D) of push-out specimens was 3.00, which is the same as the specimen used by Zhong [14]. The diameter and length of the round steel tube were 90 mm and 300 mm, respectively, and the thickness was 3 mm; the edge length and length of the square steel tube were 90 mm and 300 mm, respectively, and the thickness was 3 mm. It was sealed with a 30 mm waterproof foam board at one end of the steel tube to prevent the cement slurry from leaking out, and the tightness of the cement filling was ensured by vibrating with a vibrating table. There are twelve groups of push-out specimens, with three specimens in each group, and the details regarding of push-out specimens are shown in Table 3.

After the push-out specimens were poured, the specimens were maintained in YH-90B standard constant temperature and humidity curing box in the Civil Engineering Laboratory 101 of the Ordos Institute of Technology, as shown in Figure 3.

### 2.3. Investigation Items

After the specimens were cured for 28 days, the push-out specimens were placed in the freeze–thaw tester, and the freeze–thaw test according to GB/T 50082-2009 test standard [29]:(1)One freeze–thaw cycle time is 2–4 h, and the time used for thawing is not less than 1/4 of the entire freeze–thaw cycle time;(2)During the freezing and thawing process, the minimum and maximum temperatures in the center of the specimen are controlled at −17 °C ± 2 °C and 7 °C ± 2 °C, respectively;(3)The time is taken for each specimen to drop from 7 °C to −17 °C shall not be less than 1/2 of the freezing time, and the temperature difference between the inside and outside of the specimen shall not exceed 28 °C;(4)The transition time between freezing and thawing should not exceed 10 min.

The temperature vs. time curve of freeze–thaw cycles is shown in Figure 4.

After the push-out specimens had undergone the rated number of freeze–thaw cycles, the push-out specimens were tested with a microcomputer-controlled electronic universal testing machine, as shown in Figure 5. The load end slip and free end slip were measured by the linear variable displacement transducers (LVDT), as shown in Figure 6. The load end slip was measured by LVDT 1 and LVDT 2, while the free end slip was measured by LVDT 3 and LVDT 4. The experimental data were automatically collected by the load sensor and displacement sensor linked to the computer. The loading methods of the test were graded, and the loading speed was controlled at 0.05 kN/s. When the force–displacement curve of the testing machine display showed obvious nonlinear change, the loading speed was controlled at 0.02 kN/s.

## 3. Results and Discussion

### 3.1. Failure Mode

The failure modes of the push-out specimens are shown in Figure 7. It can be seen that the failure modes of the round and square steel tube specimens were similar. In addition, the effect of freeze–thaw cycles on the failure modes of the push-out specimens was not significant. These observations agree with the findings of Li [24] and Shen [25]. In the initial stage of loading, with the increase in the load, the displacement of the loading end increased continuously, but the displacement growth rate was slow while the displacement of the free end did not change. With the increase in the load, the displacement of the free end began to increase, but the increase rate was lower than that of the loaded end and some push-out specimens made a spitting sound at this time. When the push-out load reached the peak value, the load decreased and then continued to increase after reaching the bottom. There was no obvious buckling of the stainless steel tube or obvious damage to the concrete edge at the loading end. This agrees with the findings of Li [30].

### 3.2. Shear Bond Strength and Peak Slip

The maximum load, maximum shear bond strength, and the peak slip (slip at maximum shear bond strength) of push-out specimens after freeze–thaw cycles are shown in Table 4. The Peak slip is the average of the free end slip and the loaded end slip. The average bond strength of round steel tube specimens and square steel tube specimens is determined with Equation (1) and Equation (2), respectively:(1)$$\tau =\frac{F}{\pi dl_{a}}$$
(2)$$\tau =\frac{F}{4al_{a}}$$
where τ is the shear bond strength; F is the push-out load; d is the diameter of steel tube; *a* is the side length of square steel tube; $l_{a}$ is the bond length between steel tube and concrete.

#### 3.2.1. Shear Bond Strength after Freeze–Thaw Cycles

The shear bond strength of the round and square steel tube push-out specimens after freeze–thaw cycles are shown in Figure 8a,b. It can be seen that the shear bond strengths of these specimens all showed a downward trend with the increase in the number of freeze–thaw cycles. This agrees with the findings of Shen [25]. After 10 freeze–thaw cycles and 20 freeze–thaw cycles, the shear bond strength of the round steel tube push-out specimen with 30 MPa decreased by 20.16% and 33.33%, respectively, while that of the square steel tube push-out specimen with 30 MPa decreased by 9.88% and 50.62%, respectively. The reason is that when the concrete structure is exposed to a freeze–thaw environment, the pore solution in the capillary turns into ice, causing volume expansion and creating water pressure. When the expansion force exceeds the tensile strength of the concrete, micro cracks begin to occur and radiate to the surrounding cement paste, thereby causing damage to the concrete [31]. The freeze–thaw damage is exacerbated by the increasing number of freeze–thaw cycles [32].

In addition, with the increase in concrete strength, the shear bond strength between the steel tube and concrete also increased, while the degree of decrease in shear bond strength after freeze–thaw cycles was reduced. After 10 freeze–thaw cycles and 20 freeze–thaw cycles, the shear bond strength of the round steel tube push-out specimen with 45 MPa decreased by 17.59% and 25.01%, respectively, while that of the square steel tube push-out specimen with 45 MPa decreased by 4.55% and 43.18%, respectively. The reasons for this are as follows: on the one hand, when the water–cement ratio is large, the concrete is fully hydrated, resulting in greater porosity, resulting in more serious freeze–thaw damage; on the other hand, the greater the water–cement ratio, the lower the strength of concrete and the lower the frost resistance, resulting in shear bond performance between the concrete and steel tube after freeze–thaw cycles. Moreover, the shear bond strength of the round steel tube push-out specimen was higher than that of square steel tube push-out specimen. After 20 freeze–thaw cycles, the shear bond strength of round steel tube push-out specimen with 30 MPa/45 MPa was 0.86 MPa/1.62 MPa, while that of square steel tube push-out specimen with 30 MPa/45 MPa was 0.40 MPa/0.50 MPa.

The relationship between the shear bond strength of C30 round steel tube specimen and the number of freeze–thaw cycles was fitted as follows (*n* is the number of freeze–thaw cycles):(3)$$\tau_{p30}=-0.0215n+1.275$$

The strength conversion coefficient of the round steel tube specimen’s shear bond strength (*a*) was fitted as follows:(4)$$a=1.2729\tau_{p30}+0.504$$

The conversion coefficient of round steel tube specimen’s shear bond strength and square steel tube specimens shear bond strength (*b*) was fitted as follows:(5)$$b=0.8998\tau_{p30}-0.3071$$

The strength conversion coefficient of the square steel tube specimen’s shear bond strength (*c*) was fitted as follows:(6)$$c=0.957\tau_{s30}-0.1212$$

Table 5 lists the contrast between the experimental values ($\tau_{{}_{e}}^{}$) and the calculated values ($\tau_{{}_{c}}^{}$) of each group specimens. As shown in Table 5, the calculated values have a good correlation with the experimental values.

#### 3.2.2. Peak Slip after Freeze–Thaw Cycles

The peak slip of the round and square steel tube push-out specimens after freeze–thaw cycles are shown in Figure 9a,b. It can be seen that the peak slip of both the round and square steel tube push-out specimens presented an upward trend with the increase in the number of freeze–thaw cycles. After 10 and 20 freeze–thaw cycles, the peak slip of the round steel tube push-out specimen with 30 MPa increased by 6.89% and 11.49%, respectively, while that of the square steel tube push-out specimen with 30 MPa increased by 7.04% and 0.37%, respectively. These observations agree with the findings of Liu [12]. The reasons are as follows: on the one hand, the freeze–thaw cycle damages the bond performance of concrete and steel tube, so when external loads are applied the steel tube and concrete are more likely to slide, increasing the slip value; on the other hand, the elastic modulus of the concrete decreased after freeze–thaw cycles, resulting in a decrease in the overall stiffness of the specimen [21].

The relationship between the slip of C30 round steel tube specimen and the number of freeze–thaw cycles was fitted as follows (*n* is the number of freeze–thaw cycles):(7)$$s_{p30}=0.025n+4.3667$$

The strength conversion coefficient of the round steel tube specimen’s slip ($a^{\prime }$) was fitted as follows:(8)$$a^{\prime }=1.9579s_{p30}-4.9589$$

The conversion coefficient of round steel tube specimen’s slip and square steel tube specimen’s slip ($b^{\prime }$) were fitted as follows:(9)$$b^{\prime }=0.3816s_{p30}+1.0617$$

The strength conversion coefficient of the square steel tube specimen’s slip ($c^{\prime }$) was fitted as follows:(10)$$c^{\prime }=-91.52s_{s30}{}^{2}+511.07s_{s30}-709.45$$

Table 6 lists the contrast between the experimental values ($s_{{}_{e}}^{}$) and the calculated values ($s_{{}_{c}}^{}$) of each group of specimens. As shown in Table 6, except for the C45 square steel tube specimen, the calculated values have a good correlation with the experimental values. However, the error value of a C45 square steel tube specimen is also within 30%.

### 3.3. Load–Slip Curve

The load–slip curves of the round and square steel tube push-out specimens after different freeze–thaw cycles are shown in Figure 10 and Figure 11, respectively. There is no obvious difference between the load–slip curve shape of the round and square specimens. In addition, the shape of the load–slip curve of the steel tube push-out specimen was not significantly affected by the concrete strength and the freeze–thaw cycle. However, there were more significant differences in shear stiffness, and the shear stiffness of the specimens showed a decreasing trend after freeze–thaw cycles; this was due to the damage of the bonding surface caused by the freeze–thaw cycles, resulting in a decrease in the bond strength, and at the same time, the slippage between the steel pipe and the concrete increased; the shear stiffness of the specimens showed an increasing trend with the increase in concrete strength, which was due to the improvement of concrete strength that increased the mechanical bite force between concrete and steel tube, thereby improving the bond strength of the steel tube push-out specimen and reducing the slip between the concrete and steel tube.

The load–slip curve of the steel tube push-out specimen was roughly divided into four stages:(a)Slowed rising stage

At this stage, it mainly depended on the chemical adhesion between the steel tube and the core concrete to bear the external load. At this time, the bond performance between the core concrete and the steel tube was good, the free end did not slip, the slip at the loading end was also small, and the load–displacement curve was linear. As the number of freeze–thaw cycles increased, the interface between the steel tube and the core concrete was damaged, resulting in a decrease in chemical adhesion;

(b) Accelerated rising stage

With the continuous increase in external load, the chemical adhesion between the steel tube and the core concrete was gradually lost. The free end began to slip, and the slip between the steel tube and the core concrete increased continuously. At this time, it mainly depended on the mechanical bite force and the interface friction to bear the external load. After the freeze–thaw cycle, the damage of the freeze–thaw to the concrete and the damage of the external load to the concrete affected each other, resulting in the decrease in the bond strength and the increase in the peak slip, which reduced the slope of the load–slip curve;

(c) Descending stage

After reaching the ultimate load, the load suddenly decreased, the relative slip of the core concrete increased rapidly, and the curve had an obvious peak point. At this stage, it mainly depended on the residual mechanical bite force and the interface friction to bear the external load. As the number of freeze–thaw cycles increased, the concrete became loose, resulting in a decrease in mechanical bite force and interfacial frictional;

(d) Second rising stage

When the bond strength dropped to a certain level, it rose again. The reason for this is that the core concrete and the steel tube have a “Pinching effect” (as shown in Figure 12) [30,33], and the interfacial friction gradually increased, increasing the bond strength, and the interfacial cohesive force reached a dynamic equilibrium again, thereby improving the bond strength between the steel tube and the core concrete.

## 4. Conclusions

This paper focuses on the less-researched problem of shear bond performance at the interface of concrete-filled steel tubes with different sections after freeze–thaw cycles. The effects of the number of freeze–thaw cycles, section type of steel tube, and concrete strength on the shear bond performance of concrete-filled steel tube were investigated by experimental methods, and the results could provide a theoretical basis for the application of concrete-filled steel tube in cold regions. The main findings were as follows:(1)The shear bond strength of the round and square steel tube push-out specimens both showed a downward trend after freeze–thaw cycles. After 20 freeze–thaw cycles, the shear bond strength of the round and square steel tube push-out specimens with 30 MPa decreased by 33.33% and 50.62%, respectively;(2)The shear bond strength increased with the increase in concrete strength, while the degree of decrease in shear bond strength after freeze–thaw cycles was reduced;(3)The shear bond strength of the round steel tube push-out specimen was higher than that of the square steel tube push-out specimen. After 20 freeze–thaw cycles, the shear bond strength of the round steel tube push-out specimen with 30 MPa/45 MPa was 0.86 MPa/1.62 MPa, while that of square steel tube push-out specimen with 30 MPa/45 MPa was 0.40 MPa/0.50 MPa;(4)The shear bond strength and peak slip of the push-out specimen after freeze–thaw cycles calculation formula was fitted;(5)The peak slip of the round and square steel tube push-out specimens presented an upward trend with an increase in the number of freeze–thaw cycles. After 20 freeze–thaw cycles, the peak slip of the round and square steel tube push-out specimens with 30 MPa increased by 11.49% and 0.37%, respectively;(6)The shape of the load–slip curve of the steel tube push-out specimens was not significantly affected by the steel tube section type, the concrete strength, and the freeze-thaw cycle; the load–slip curve was roughly divided into the slow rising stage, the accelerated rising stage, the descending stage, and the second rising stage.

According to the relationship between the number of rapid freeze–thaw cycles in literature [34] and the actual number of freeze–thaw cycles, it can be seen that for specimens with 30 MPa, 20 rapid freeze–thaw cycles are about 75 actual freeze–thaw cycles; for the specimen with 45 MPa, 20 rapid freeze–thaw cycles are about 50 actual freeze–thaw cycles. The frost resistance of concrete-filled steel tube specimens was poor. Therefore, the frost resistance of concrete should be improved by adding air entraining agent and concrete admixture to improve the shear bond performance of concrete-filled steel tube specimen after freeze–thaw cycles.

## Figures and Tables

**Figure 1 materials-15-07233-f001:**
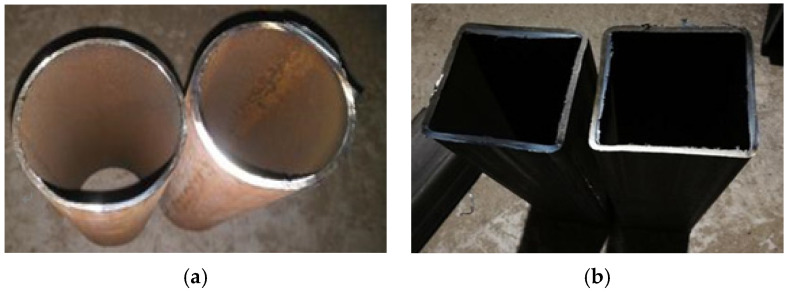
Steel tube: (**a**) Round steel tube; (**b**) Square steel tube.

**Figure 2 materials-15-07233-f002:**
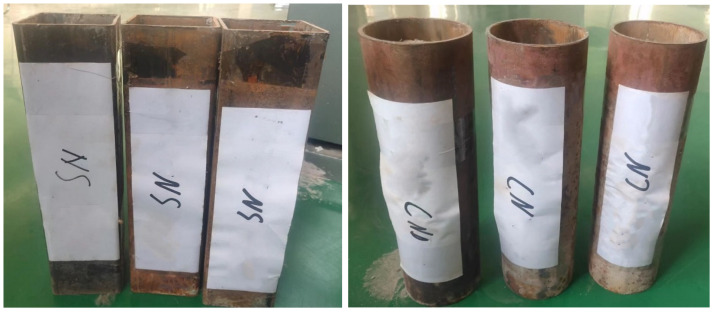
Concrete-filled steel tube push-out specimens.

**Figure 3 materials-15-07233-f003:**
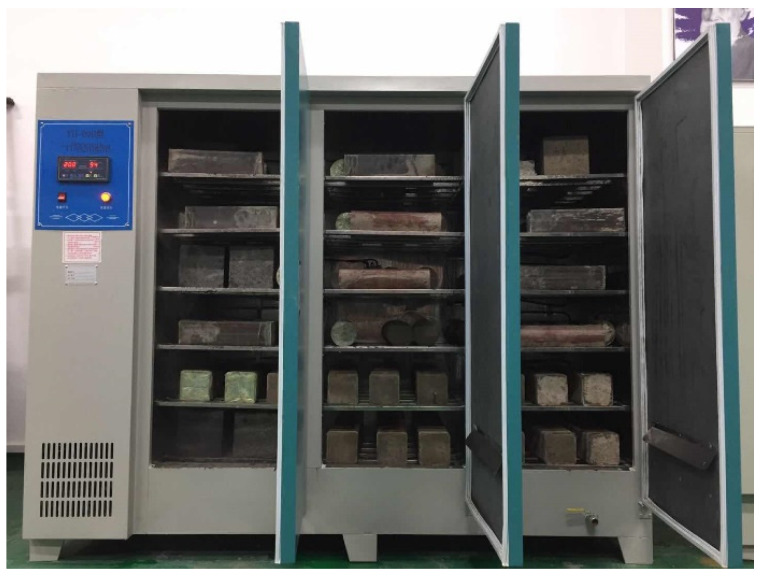
Constant temperature and humidity curing box.

**Figure 4 materials-15-07233-f004:**
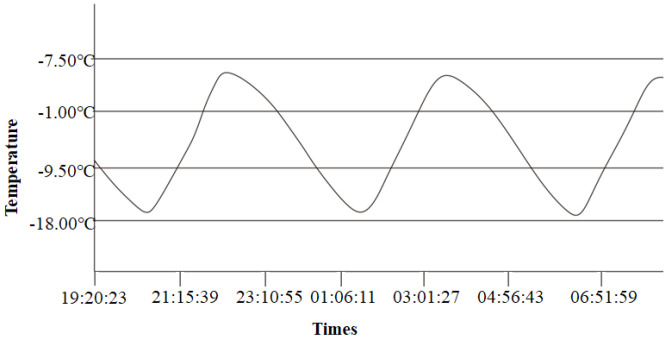
Temperature vs. time curve of freeze–thaw cycles.

**Figure 5 materials-15-07233-f005:**
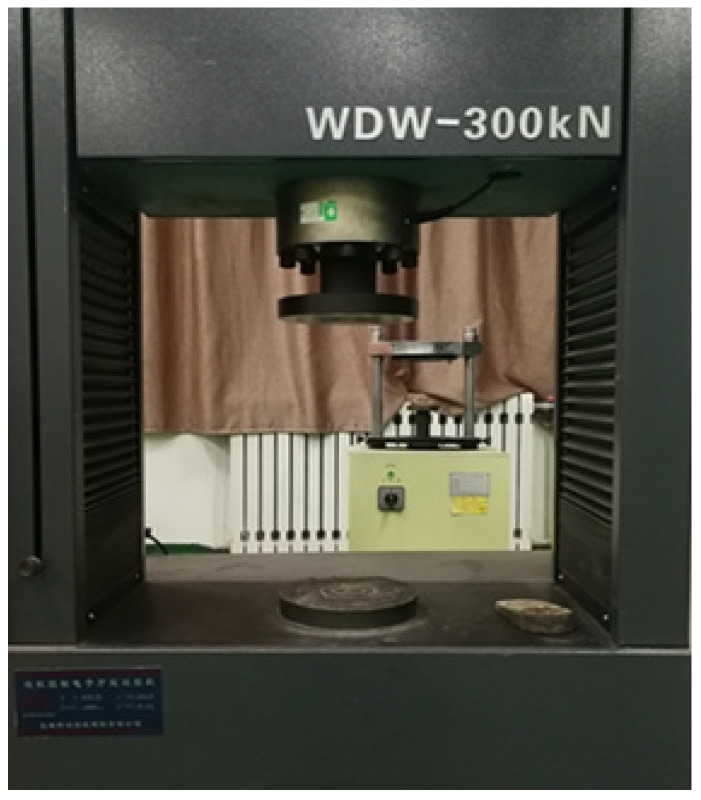
Electronic universal testing machine.

**Figure 6 materials-15-07233-f006:**
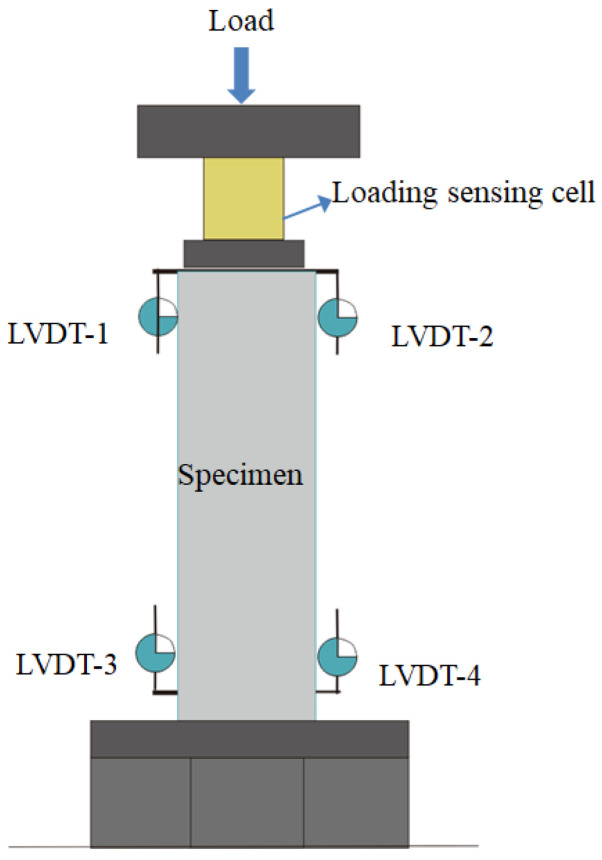
Schematic diagram of specimen loading.

**Figure 7 materials-15-07233-f007:**
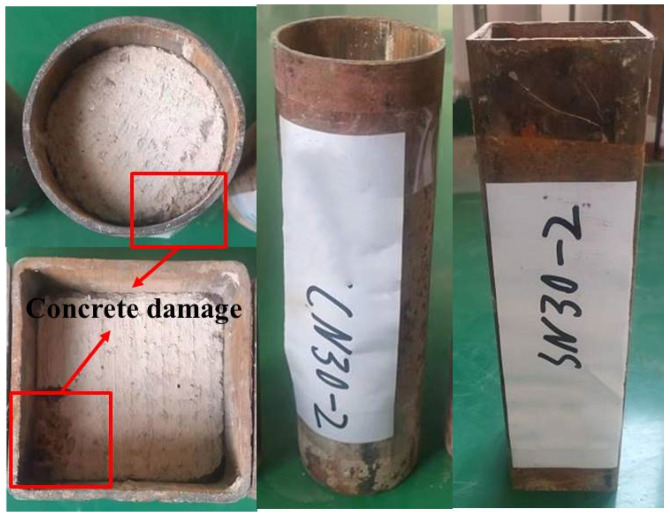
Failure mode of push-out specimens.

**Figure 8 materials-15-07233-f008:**
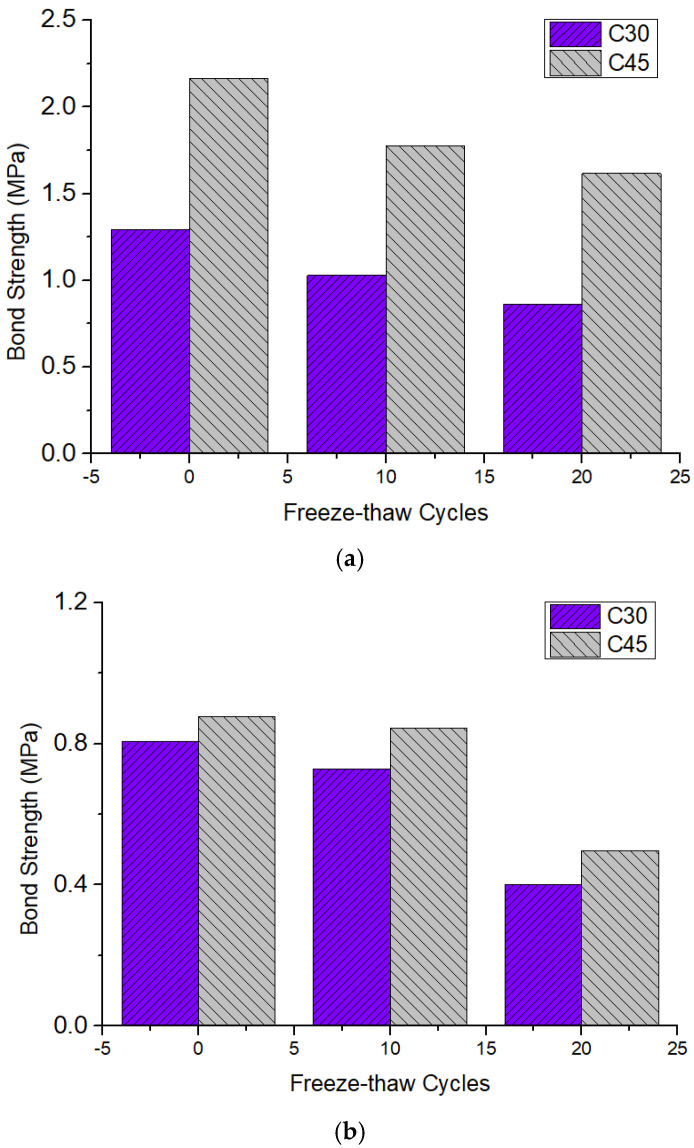
Failure mode of push−out specimens: (**a**) Round steel tube push-out specimen; (**b**) Square steel tube push-out specimen (the data in the figure are averaged).

**Figure 9 materials-15-07233-f009:**
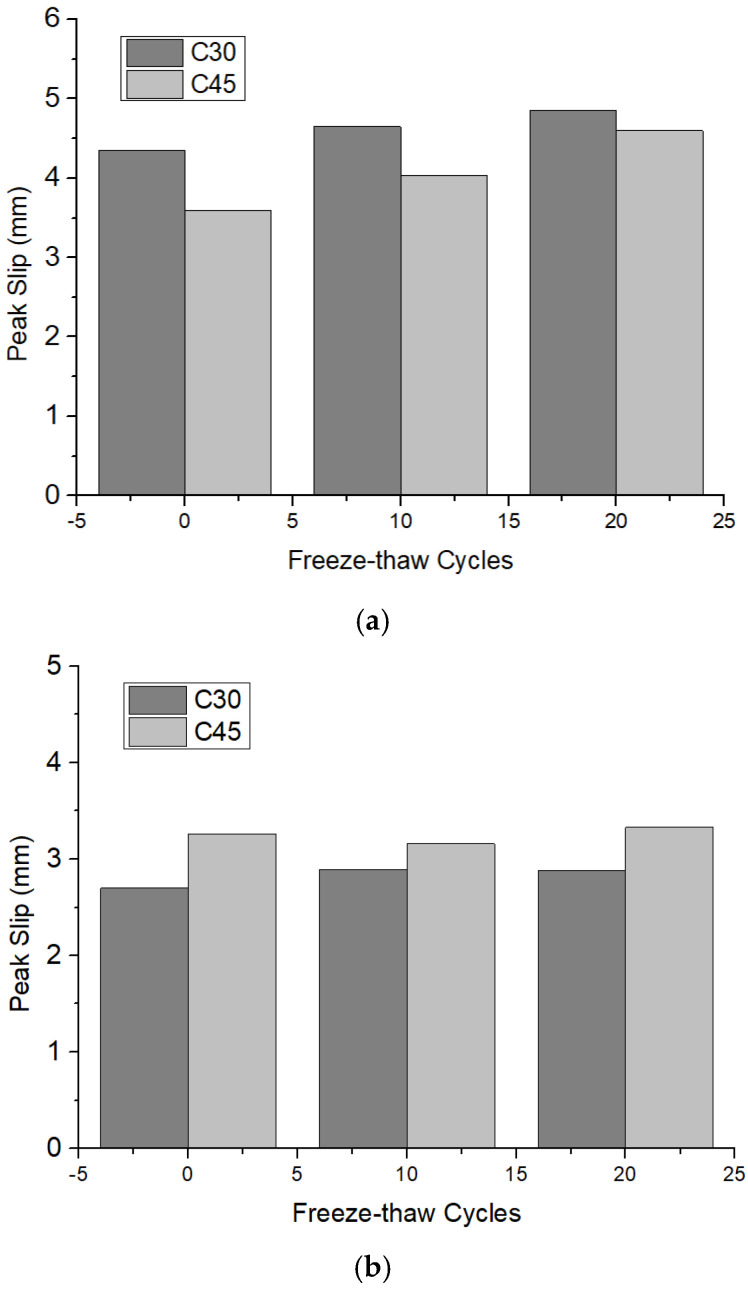
Peak slip of push−out specimen after freeze–thaw cycles: (**a**) Round steel tube push-out specimen; (**b**) Square steel tube push-out specimen (the data in the figure are averaged).

**Figure 10 materials-15-07233-f010:**
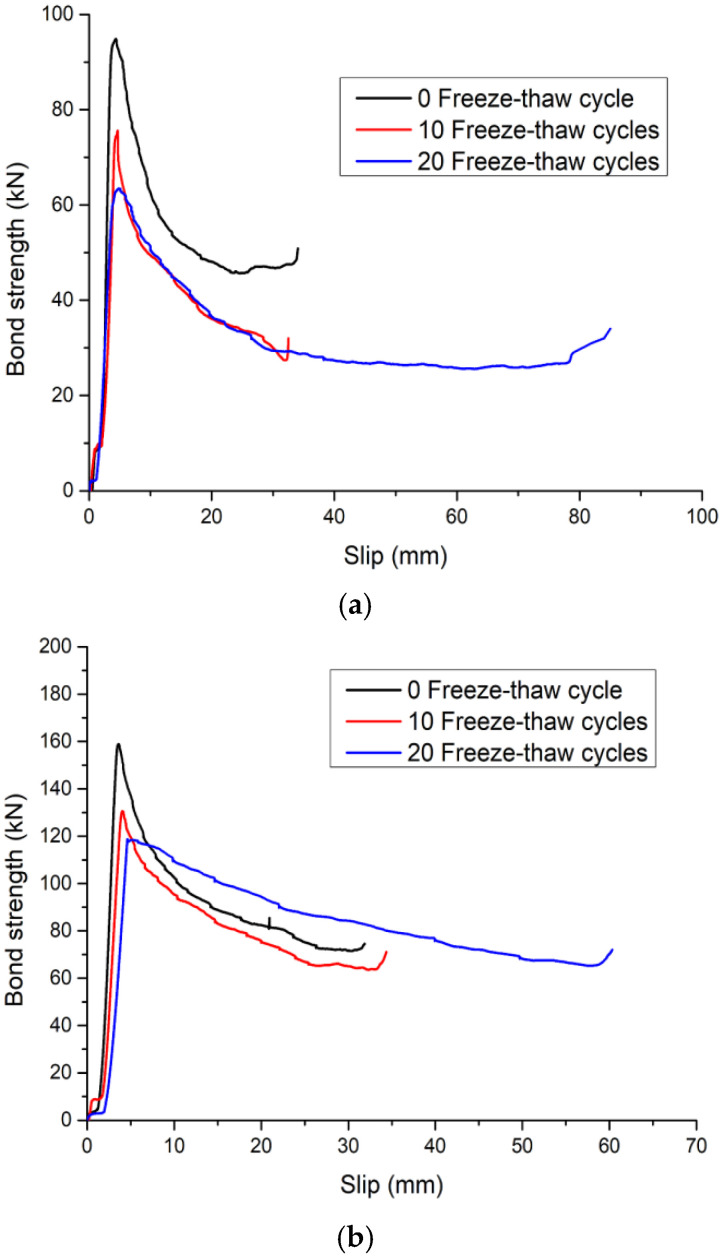
Load–slip curve of round steel tube push-out specimen: (**a**) C30; (**b**) C45 (the data in the figures are averaged).

**Figure 11 materials-15-07233-f011:**
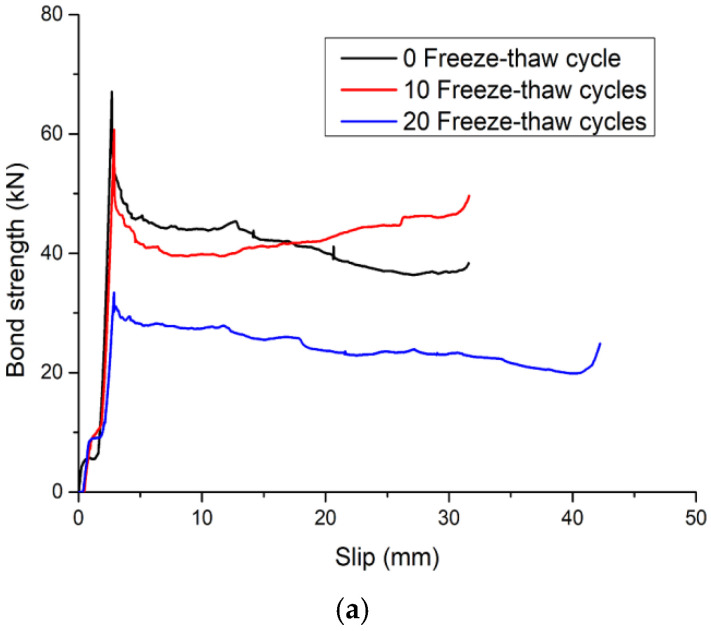
Load–slip curve of square steel tube push-out specimen: (**a**) C30; (**b**) C45 (the data in the figures are averaged).

**Figure 12 materials-15-07233-f012:**
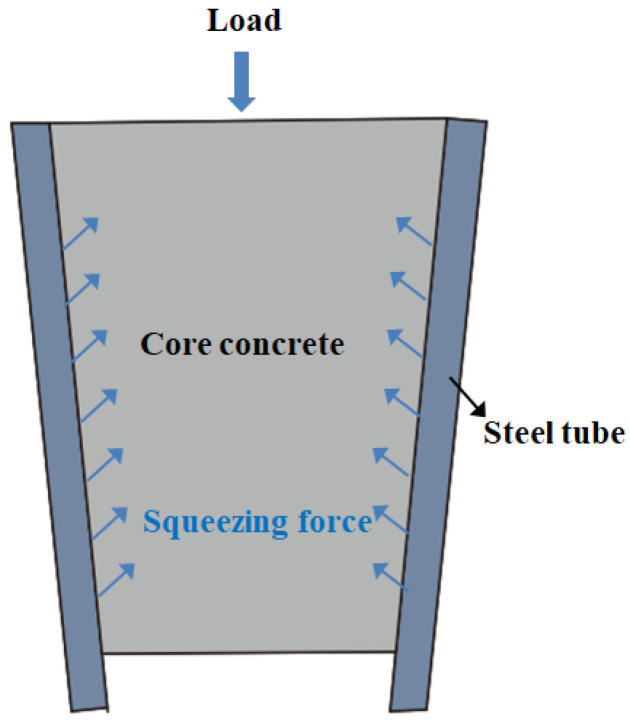
Pinching effect diagram.

**Table 1 materials-15-07233-t001:** Performance of cement.

Type	Initial Setting Time/min	Final Setting Time/min	Compressive Strength/MPa		Flexural Strength/MPa	
			3-Day	28-Day	3-Day	28-Day
P.O 42.5	178	251	24.5	48.8	5.2	8.9

**Table 2 materials-15-07233-t002:** Mix proportions of concrete (kg m^−3^).

Property	Concrete (30 MPa)	Concrete (45 MPa)
Cement	415.9	503.7
Water	208.0	208.0
Coarse aggregate	1141.6	1188.4
Fine aggregate	562.3	542.84

**Table 3 materials-15-07233-t003:** Details regarding of push-out specimens.

Specimen	Concrete Strength	Steel Tube Type	Freeze–Thaw Cycles
P-30-0	C30	Round steel tube	0
P-30-10	C30	Round steel tube	10
P-30-20	C30	Round steel tube	20
S-30-0	C30	Square steel tube	0
S-30-10	C30	Square steel tube	10
S-30-20	C30	Square steel tube	20
P-45-0	C45	Round steel tube	0
P-45-10	C45	Round steel tube	10
P-45-20	C45	Round steel tube	20
S-45-0	C45	Square steel tube	0
S-45-10	C45	Square steel tube	10
S-45-20	C45	Square steel tube	20

**Table 4 materials-15-07233-t004:** Shear bond strength and peak slip of push-out specimens.

Specimens	Shear Bond Strength (MPa)	Average Shear Bond Strength (MPa)	Standard Deviation/Variation Coefficient	Peak Slip (mm)	Average Peak Slip (mm)	Standard Deviation/Variation Coefficient
P-30-0-1	1.31	1.29	0.0556/0.0681	4.47	4.35	0.1250/0.1531
P-30-0-2	1.21			4.41		
P-30-0-3	1.34			4.18		
P-30-10-1	1.12	1.03	0.0700/0.0862	4.76	4.65	0.0818/0.1002
P-30-10-2	1.01			4.61		
P-30-10-3	0.95			4.57		
P-30-20-1	0.95	0.86	0.0698/0.0854	4.94	4.85	0.0896/0.1097
P-30-20-2	0.78			4.73		
P-30-20-3	0.85			4.89		
S-30-0-1	0.88	0.81	0.0510/0.0625	2.87	2.70	0.2042/0.2501
S-30-0-2	0.79			2.41		
S-30-0-3	0.76			2.81		
S-30-10-1	0.71	0.73	0.0556/0.0681	2.81	2.89	0.0741/0.0907
S-30-10-2	0.68			2.99		
S-30-10-3	0.81			2.88		
S-30-20-1	0.58	0.40	0.1266/0.1550	2.78	2.88	0.0818/0.1002
S-30-20-2	0.29			2.87		
S-30-20-3	0.34			2.98		
P-45-0-1	2.45	2.16	0.2089/0.2558	3.68	3.60	0.0580/0.0709
P-45-0-2	1.98			3.54		
P-45-0-3	2.04			3.59		
P-45-10-1	1.89	1.78	0.0900/0.1212	4.12	4.04	0.0655/0.0802
P-45-10-2	1.65			3.96		
P-45-10-3	1.8			4.05		
P-45-20-1	1.7	1.62	0.0556/0.0681	4.78	4.60	0.14727/0.18037
P-45-20-2	1.57			4.58		
P-45-20-3	1.6			4.42		
S-45-0-1	0.97	0.88	0.0736/0.0902	3.28	3.26	0.0624/0.0764
S-45-0-2	0.89			3.33		
S-45-0-3	0.79			3.18		
S-45-10-1	0.91	0.84	0.0613/0.0751	3.28	3.16	0.1123/0.1375
S-45-10-2	0.84			3.01		
S-45-10-3	0.76			3.19		
S-45-20-1	0.56	0.50	0.0419/0.0513	3.58	3.33	0.1926/0.2359
S-45-20-2	0.46			3.11		
S-45-20-3	0.49			3.31		

**Table 5 materials-15-07233-t005:** Contrast between bond strength experimental values ($\tau_{{}_{e}}^{}$) and bond strength calculated values ($\tau_{{}_{c}}^{}$).

Specimens	$\tau_{{}_{e}}^{}$ (MPa)	$\tau_{{}_{c}}^{}$ (MPa)	$\tau_{{}_{e}}^{}/\tau_{c}^{}$
P-30-0	1.29	1.28	1.01
P-30-10	1.03	1.06	0.97
P-30-20	0.86	0.5	1.72
S-30-0	0.81	0.84	0.96
S-30-10	0.73	0.66	1.11
S-30-20	0.40	0.45	0.89
P-45-0	2.16	2.13	1.01
P-45-10	1.78	1.85	0.96
P-45-20	1.62	1.58	1.03
S-45-0	0.88	0.93	0.95
S-45-10	0.84	0.74	1.14
S-45-20	0.50	0.55	0.91

**Table 6 materials-15-07233-t006:** Contrast between bond strength experimental values ($s_{{}_{e}}^{}$) and bond strength calculated values ($s_{{}_{c}}^{}$).

Specimens	$s_{{}_{e}}^{}$ (MPa)	$s_{{}_{c}}^{}$ (MPa)	$s_{{}_{e}}^{}/s_{{}_{c}}^{}$
P-30-0	4.35	4.37	0.99
P-30-10	4.65	4.62	1.01
P-30-20	4.85	4.87	0.99
S-30-0	2.70	2.73	0.99
S-30-10	2.89	2.82	1.02
S-30-20	2.88	2.92	0.99
P-45-0	3.60	3.59	1.01
P-45-10	4.04	4.08	0.99
P-45-20	4.60	4.57	1.01
S-45-0	3.26	3.66	0.89
S-45-10	3.16	3.951	0.80
S-45-20	3.33	2.57	1.30

## Data Availability

The data used to support the findings of this study are included in the paper.

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
