# Peer review of "Shear Bond Performance at Interface of Concrete-Filled Steel Tube after Freeze–Thaw Cycles"

_materials, 2022, doi:10.3390/ma15207233_

Round 1
Reviewer 1 Report
After reviewing the paper, I noticed certain shortcomings, which I give below:
Correct the names of images 6 and 7.
By increasing the number of freezing and thawing cycles, a clear trend of decreasing shear strength can be seen. Why the tests were completed with 20 freeze-thaw cycles. Why is that number of cycles not higher? At what number of freezing and thawing cycles does the bonds break or the shear strength disappears.
Likewise, the number of cycles should be referred to in the conclusions. What about the real number of cycles during the life of the construction? An obvious trend of decreasing shear strength in freezing and thawing cycles leads to its yielding. Ultimately, how to take this effect into account when designing structures.
Reviewer 2 Report
The paper deals with the interesting and applicable subject of strength behavior of the concrete filled steel tubes. Generally the paper is correct, but in my opinion the methodology and results description needs an improvement:
1. Please add the quantitative results in the abstract.
2. The studied samples are not described. Please show the dimensions of the tubes and please describe how the correctness and tightness of cement filling was verified (it strictly influences the suggested “pinching effect”). Also the material properties (especially thermal) for tubes must be shown. The thermal properties of used concrete should be also shown.
3. Please add the description of mechanical testing - were the samples tested after the conditioning in room temperature or just after freezing?
4. Please show the course of freeze -thaw cycles (temperature vs. time curve is necessary).
5. Please show the statistical analysis of achieved results (scatter of results, deviation, mean, etc.).
6. The presentation of tubes failure (fig. 8) is not clear – please show the photos of the magnified “scratches”. They look like the results of removing the concrete filling – please discuss. Are those results repeatable?
7. The conclusions are too overall. Please add the conditions of the experiments (freeze-thaw cycles specification, materials properties) on the results. Please add the comparative analysis of the results and please add the literature examples to validate the correctness of the applied methods and achieved results.
Reviewer 3 Report
Abstract should be improved. Please mention the main problem which leads to the necessity of these investigations. Please mention the methods have been used to determine the shear bond strength as well as the parameters for freeze-thaw cycles. The abstract should not contain such a detailed description of the results.
Introduction: please mention which are the main mechanisms responsible for the bond performance between the steel pipe and the concrete? You mention the interfacial bond strength between the steel tube and the concrete. Please describe the nature of the “interfacial bond”.
Please mention some reference values for the “height-to-diameter ratio” from literature, in order to justify the selected dimensions for the tested samples.
Figure 5 is not necessary.
Please adapt the text from Figures 6 and 7.
Are the values from table 4 the results obtained just from one measurement or there are average values from both samples, each specimen type? There are 24 samples mentioned. The same question for the rest of figures starting with figure 9. Standard deviation is necessary to be mentioned.
Conclusions, point 2: can you make any difference between the behavior observed depending on the shape of the tubes (round, square)?
Please formulate better in the conclusion which is the novelty of the results in comparison with the literature?
Round 2
Reviewer 1 Report
The authors have corrected the paper according to the reviewers' requirements and the paper can be published.
Reviewer 2 Report
All my comments are considered. In my opinion the paper can be accepted,
Reviewer 3 Report
The authors improved the manuscript according to the made comments.